# Fabrication of Polystyrene/AlOOH Hybrid Material for Pb(II) Decontamination from Wastewater: Isotherm, Kinetic, and Thermodynamic Studies

**Rajeev Kumar** 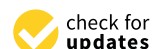

Department of Environmental Sciences, Faculty of Meteorology, Environment and Arid Land Agriculture, King Abdulaziz University, Jeddah 21589, Saudi Arabia; rsingh@kau.edu.sa or olifiaraju@gmail.com

**Abstract:** The nanomaterials' toxicity to aquatic life is a big issue due to improper handling or incomplete separation after use. The immobilization of the nanomaterials in the polymeric matrix could be a practical approach to developing an efficient hybrid composite for wastewater purification. In this study, AlOOH nanoparticles were immobilized in the polystyrene polymeric matrix to prepare an effective adsorbent to scavenge the Pb(II) from the aqueous solution. The synthesized polystyrene/AlOOH (PS/AlOOH) hybrid was characterized using microscopic techniques coupled with elemental mapping and EDX, X-ray diffraction, and a furrier-transformed infrared spectrometer. The results revealed that the Pb(II) adsorption onto the polystyrene/AlOOH composite depends on the solution pH, the Pb(II) concentrations in the solution, the adsorption time, and the solute temperature. The maximum scavenging of Pb(II) occurs at pH 6 in 90 min. The adsorption of Pb(II) onto PS/AlOOH decreases from 97.7% to 58.5% with the increase in the Pb(II) concentration from 20 mg g$^{-1}$ to 100 mg g$^{-1}$. The kinetics and isotherm modeling demonstrated that Pb(II) adsorption is well suited for the pseudo-second-order kinetics and Toth isotherm models, suggesting that the chemisorption occurs at the heterogeneous surface of PS/AlOOH. The PS/AlOOH composite could be used multiple times without a significant loss in the adsorption efficiency. These results demonstrated that the polystyrene/AlOOH composite is an effective material for the purification of wastewater and can be used on a large scale.

**Keywords:** polystyrene/AlOOH hybrid; water purification; lead removal; adsorption

## 1. Introduction

Water pollution from industrial activities is a severe environmental problem that encircles several organic and inorganic pollutants. However, organic pollutants could be biodegraded, while metals accumulate in living organisms, are non-biodegradable, and have adverse effects. Lead is widely used in battery manufacturing, printing, and painting and is a significant source of lead pollution. Lead is risky in low doses (the WHO recommended amount is 10 ppb), as it accumulates in body parts such as the brain, liver, bones, and kidneys [1]. Therefore, influential technology for Pb(II) scavenging from wastewater is needed to protect the environment.

In recent decades, chemical precipitation, phytoremediation, ion exchange, membrane-based methods, and many other techniques have been explored to scavenge contaminants [2]. The applications of these methods are limited or have some shortcomings, including the high energy consumption, money investment, and large volumes of sludge. Among them, adsorption is one of the most promising technologies for removing Pb(II) because of the ease of operation and comparatively low application cost. In the last few years, a different class of materials has been studied for the scavenging of organic and inorganic pollutants, such as carbons [3], agricultural waste [4,5], polymeric adsorbents [6], metal oxides [1,7], etc. Very few of them expressed good adsorption properties. Inorganic solids such as AlOOH, silica, iron oxides, etc. are well-characterized adsorbents widely used to remove charged metallic

ions. AlOOH is well known for its environment-friendly nature. Various natural materials such as mica, kaolin, kyanite, and so forth have been investigated for the decontamination of the pollutants in wastewater [8,9]. The straightforward synthesis, high stability, high surface area, and low cost of AlOOH make it one of the most demanding materials for decontaminating heavy metals from wastewater. Keshtkar [10] investigated the adsorption capability of γ-alumina for the Ni(II) and reported 99.6% scavenging. The recovery of the Cd(II) using γ-alumina/β-cyclodextrin has been investigated by Esfanjani [11]. The authors declare that experimental conditions are essential for Cd(II) recovery, and 94.15% of Cd(II) was recovered at pH 9. The existing –OH groups on the alumina surface are the key factor for the high adsorption of the metal ions [12]. The hydrophilic nature and fine or nano size of AlOOH particles are difficult to handle in an aqueous medium. During the fixed bed column study or high-pressure flow system experiments, nanoparticles that are not correctly fixed on the bed may show inconsistent results. The problem can be fixed by preparing the hybrid adsorbents by irreversibly dispersing inorganic nanoparticles within different polymeric matrices [13].

Polystyrene-based polymeric ion exchangers such as Amberlite and Dowex have been widely applied to decontaminate toxic metallic pollutants from aqueous solutions for environmental pollution control. [14,15]. The high thermal stability, reusability, and recycling nature of polystyrene-based materials make them the most promising material for water purification applications. Heavy metal removal by polystyrene-based ion-exchangers is usually accomplished via ion exchange, chelation, or complex formation with the metal ions [16]. Few recent studies have been published on the polystyrene microplastic adsorption capabilities of metallic contaminants from the aquatic medium [17,18]. Thus, polystyrene as a polymer support for preparing an organic–inorganic hybrid adsorbent could be a good option for removing toxic metal ions from industrial wastewater.

The present work prepared a polystyrene/AlOOH (PS/AlOOH) hybrid adsorbent using simple immobilizing AlOOH nanoparticles in the polystyrene matrix. The application of the PS/AlOOH hybrid was investigated for the scavenging of Pb(II) from an aqueous medium. The adsorption kinetics, isotherm models, and thermodynamics studies were performed to identify the Pb(II) adsorption mechanism onto the PS/AlOOH hybrid.

## 2. Materials and Method

### 2.1. Materials

Lead nitrate $Pb(NO_3)_2$ (98.2%) and aluminum nitrate $Al(NO_3)_3 \cdot 9H_2O$ (98%) were bought from Panreac Quimica S.A. Acetone (99.5%) was obtained from PanReac Applichem, ITW reagents. Polystyrene beads were supplied by BDH Ltd. A fixed amount of lead nitrate was dissolved in de-ionized water to make the standard Pb(II) solution.

### 2.2. Instrumentation

Scanning electron microscopy (SEM) images of AlOOH and PS/AlOOH were recorded on JSM7600F, JEOL. The X-ray diffraction (XRD) patterns of AlOOH, PS/AlOOH, and Pb(II)-adsorbed PS/AlOOH were recorded on ALTIMA-IV, RIGAKU spectroscopy. The transmission electron microscopy (TEM) imaginings of the PS/AlOOH hybrid adsorbent were recorded on a JEOL 200CX electron microscope. The Fourier transform infrared spectroscopy (FTIR) spectra of PS/AlOOH and Pb(II)-adsorbed PS/AlOOH in the wavelength range of 400–4000 $cm^{-1}$ were acquired on a 100 FTIR Perkin Elmer Spectrum spectrophotometer. The amount of Pb(II) in the solution was analyzed by a DR6000 UV-visible spectrophotometer using a HACH LCK 306 standard kit.

### 2.3. Preparation of Polystyrene/AlOOH Hybrid

Herein, a sol-gel method was used to synthesize AlOOH from aluminum nitrate as a source of aluminum. Initially, 10 g of aluminum nitrate was dissolved in 100 mL of de-ionized water. After that, 9 mL of ammonia solution (28%) was added dropwise under

stirring. A white precipitate was formed and left to stir for 16 h. The white product was thoroughly washed with de-ionized water and ethanol and dried at 105 °C for 18 h.

The PS/AlOOH hybrid adsorbent was synthesized using the modified method reported elsewhere [19] by mixing the PS and AlOOH in an equal ratio (W/W). Initially, 2.5 g of AlOOH was dispersed in 75 mL of acetone and sonicated for 30 min. After that, 2.5 g of PS beads were mixed in an acetone solution and stirred for 24 h. The PS beads became soft and formed a gel-like structure, and AlOOH was blended with the PS. After that, the PS-blended AlOOH was isolated from the acetone and dried at 70 °C for 4 h to evaporate the acetone. The obtained bulk PS/AlOOH hybrid was ground to make the fine powder and sieved (50–150 mesh size). The prepared PS/AlOOH hybrid was used for the Pb(II) adsorption studies.

### 2.4. Batch Adsorption Studies

The adsorption efficiency of PS/AlOOH for Pb(II) was explored in the batch experiments. A fixed mass of PS/AlOOH (0.05 g) was dispersed in the 20 mL Pb(II) solutions of known concentration ranges from 20 to 100 mg/L at a continual shaking speed of 200 rpm in a water bath shaker at 32 °C. The experiment was performed in the interaction time range from 0 to 150 min for the equilibrium time analysis by taking 0.05 g of PS/ALOOH mixed with 20 mL of Pb(II) solution of 100 mg $L^{-1}$ concentration. The lead ions present as Pb(II) at a low concentration and a pH of less than 6. Therefore, the solution pH was adjusted from 2.0 to 6.0 to avoid the hydrolysis product formation of lead, and 0.01 M HCl and/or 0.1 M NaOH were used to set the desired pH. In contrast, the formation of Pb(II) hydrolysis products, such as $PbOH^+$ and $Pb(OH)_2$, occurs at a pH greater than 6.0, which might lead to its precipitation [20]. The Pb(II) concentration in the supernatant solution was analyzed using a DR-6000 spectrophotometer. The Pb(II) uptake capacity per unit mass of PS/ALOOH at any time was estimated as follows:

$$q_t = (C_0 - C_e)V/M \tag{1}$$

where $q_t$ is the Pb(II) adsorption capacity of PS/ALOOH in (mg g$^{-1}$) at time $t$. $C_0$ (initial) and $C_e$ (equilibrium) are the Pb(II) concentrations in mg $L^{-1}$. V and M represent the used Pb(II) solution volume (L) and PS/ALOOH mass (g), respectively.

### 2.5. Desorption Studies

The spent PS/ALOOH adsorbent was regenerated by the chemical treatment method to make the process more economical. For the regeneration, 0.05 g of completely exhausted PS/AlOOH was mixed with 20 mL of an eluent. Water, 0.1N NaCl, and 0.1M HCl were used as the eluent. After the Pb(II) desorption in the eluent, the used PS/AlOOH was washed with water and used to scavenge the Pb(II). This cycle was repeated three times.

## 3. Results and Discussion

### 3.1. Synthesis and Characterization PS/AlOOH Hybrid Adsorbent

The sol-del method successfully synthesized nanosized AlOOH particles and immobilized them in the polystyrene (PS) matrix. The hypothesis for AlOOH immobilization with PS was to recover the AlOOH nanoparticles after adsorption. The immobilization of the AlOOH nanoparticles in the PS may slow down the motion of the polymeric chain and form an interfacial layer around the AlOOH nanoparticles, which prevents the agglomeration of the hydrophilic AlOOH nanoparticles [19]. A schematic diagram for preparing the PS/AlOOH hybrid is displayed in Figure 1. The synthesized AlOOH and PS/AlOOH hybrid adsorbent was characterized using several instrumental analyses such as XRD, SEM, TEM, elemental mapping, EDX, and FTIR.

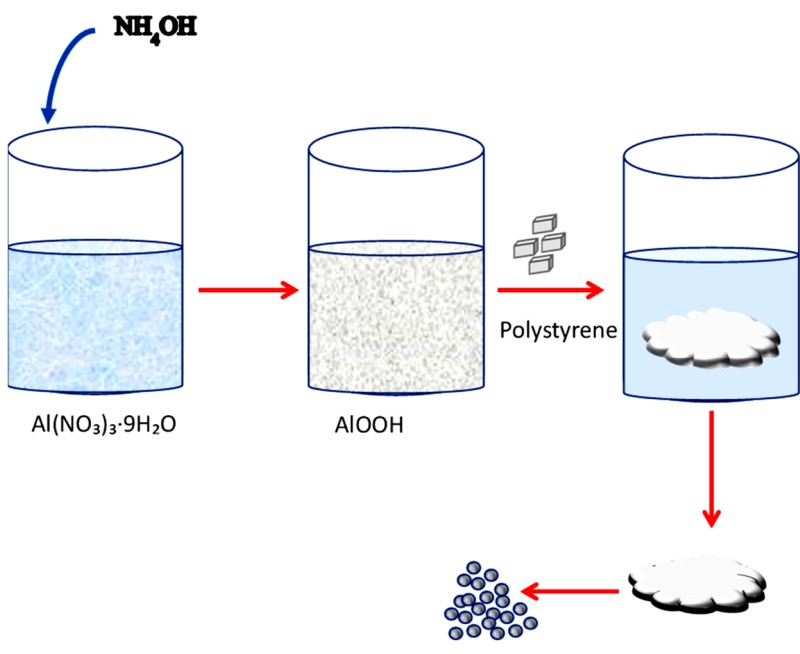

**Figure 1.** A schematic diagram for preparing the PS/AlOOH hybrid adsorbent.

The XRD patterns of AlOOH, PS/AlOOH, and Pb(II)-adsorbed PS/AlOOH are presented in Figure 2. The XRD pattern of the AlOOH shows the major characteristic peaks at 18.69° (001), 20.17° (020), 27.81° (120), 40.83° (201), 53.17° (024), and 62.21° (071), indicating the successful synthesis of the AlOOH with slight impurities of the $Al_2O_3$, which is formed due to the dehydration of few AlOOH particles. The crystallite size of AlOOH was observed in the range of 20.8 to 48.10 nm. The XRD pattern of the PS/AlOOH hybrid shows a broad peak indicating the amorphous nature of the prepared hybrid material. A similar pattern for the Pb(II)-adsorbed PS/$Al_2O_3$ hybrid was observed. The amorphous nature of the PS/AlOOH hybrid is due to the polystyrene, a highly amorphous polymer [21].

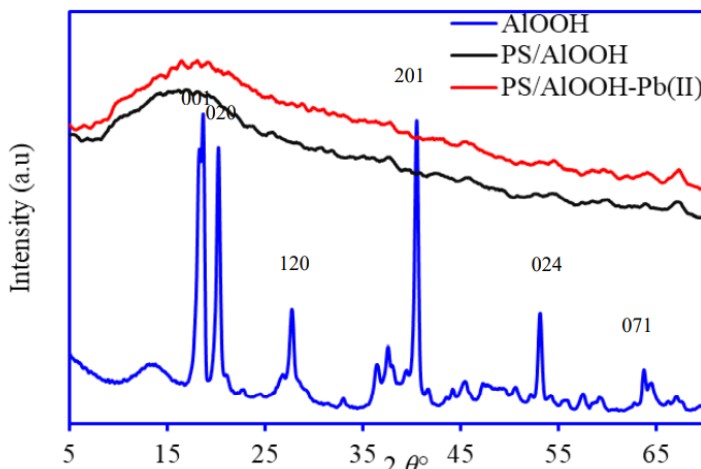

**Figure 2.** XRD pattern of the AlOOH, PS/AlOOH, and Pb(II)-adsorbed PS/AlOOH.

The surface morphologies of the AlOOH, PS/AlOOH, and Pb(II)-adsorbed PS/$Al_2O$ were recorded on the scanning electron microscopy, and the SEM images are shown in Figure 3. The SEM image of the AlOOH (Figure 3a) shows the agglomerated particles of various shapes and sizes. The AlOOH particles immobilized in the PS matrix show a flake-like morphology (Figure 3b), and embedded AlOOH particles are visible. However,

Pb(II)-adsorbed PS/AlOOH (Figure 3c) shows a similar morphology because adsorbed Pb(II) cannot be seen in the microscopy. TEM analysis further explored the morphology and interface between PS and AlOOH. TEM images of the PS/AlOOH (Figure 4a) show that the polymeric matrix contains the AlOOH nanoparticle with a porous morphology. The AlOOH particles are well distributed in the PS polymeric network (Figure 4b).

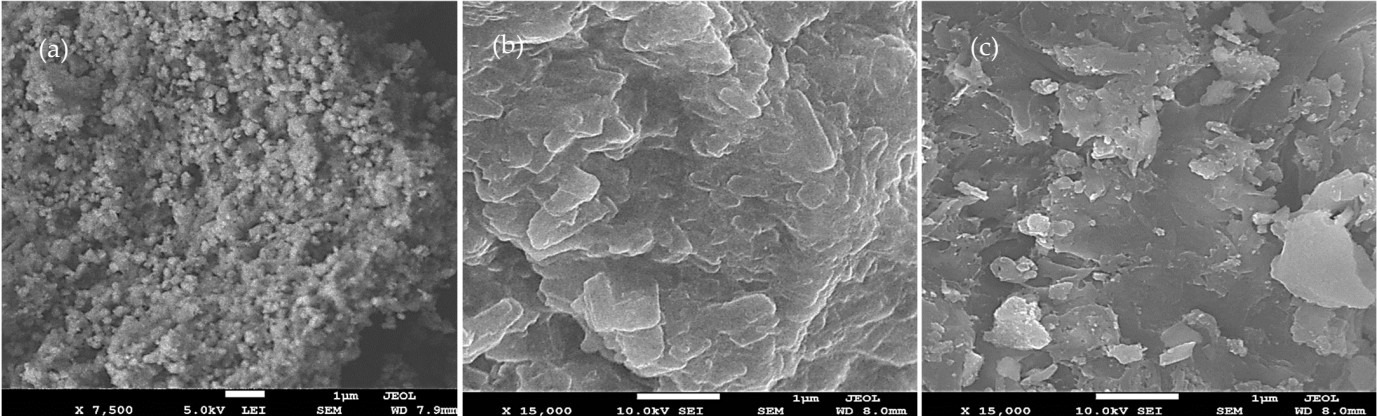

**Figure 3.** SEM images of (**a**) AlOOH, (**b**) PS/AlOOH, and (**c**) Pb(II)-adsorbed PS/AlOOH.

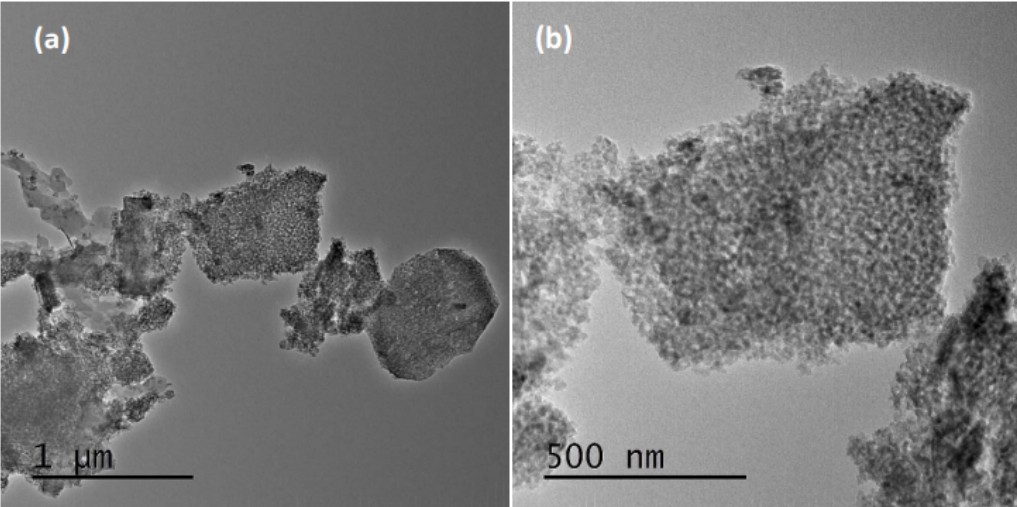

**Figure 4.** TEM images of the PS/AlOOH hybrid adsorbent.

EDX analysis was performed to confirm the Pb(II) adsorption onto the PS/AlOOH surface, and the EDX spectra are presented in Figure 5a. The EDX analysis shows the existence of the C, O, Al, and Pb elements, with 67.99, 21.51, 8.63, and 1.87% (wt%), respectively. These results reveal the successful adsorption of Pb(II) onto the PS/AlOOH surface. Moreover, to observe the distribution of the elements, elemental mapping was performed, and the mapping images are shown in Figure 5b–f. The mapping images show the good distribution of each element, indicating the good mixing of the PS and AlOOH. The elemental mapping shows the presence of carbon (Figure 5b), aluminum (Figure 5c), oxygen (Figure 5d), and surfaced adsorbed lead (Figure 5e), while Figure 5f shows all the observed elements in the Pb(II)-adsorbed PS/AlOOH hybrid adsorbent. The mapping of the Pb(II)-adsorbed PS/AlOOH shows the good distribution of the Pb(II) particles all over the surface, indicating that all the active sites contributed to the adsorption process.

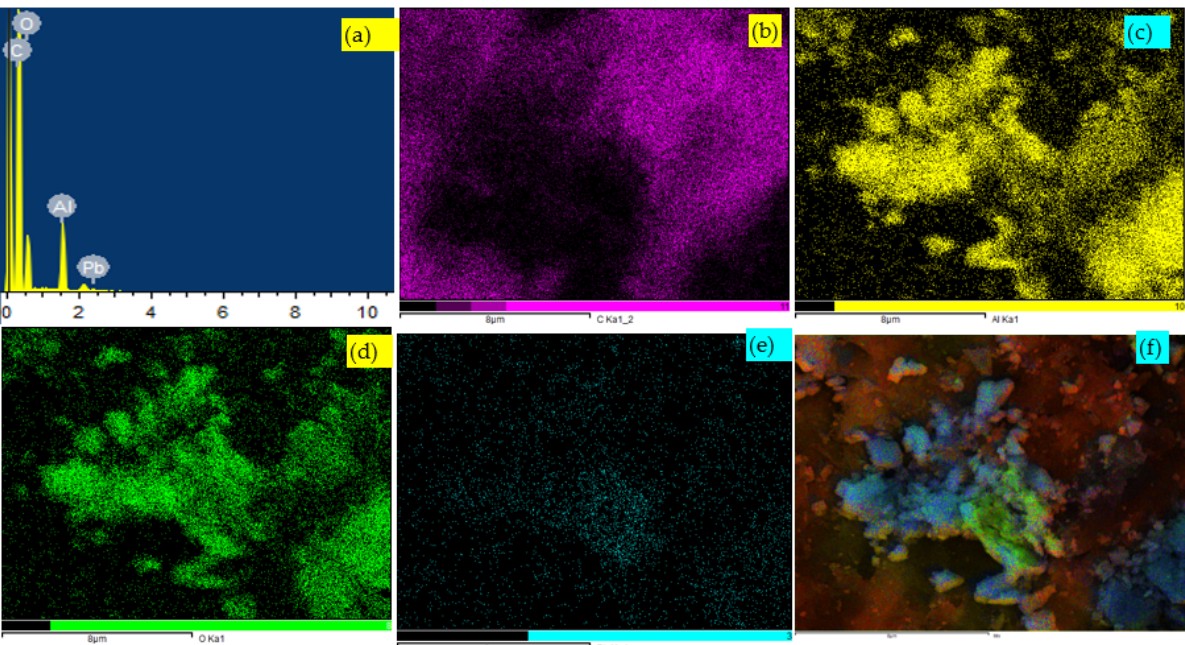

**Figure 5.** (**a**) EDX analysis and elemental mapping of the Pb-adsorbed PS/AlOOH hybrid adsorbent; (**b**) C, (**c**) Al, (**d**) O, (**e**) Pb, and (**f**) mixed elements.

FTIR analysis of the PS/AlOOH after Pb(II) adsorption was performed to find the presence of the functional groups and their interaction with Pb(II). Figure 6 illustrates the FTIR spectrum of PS/AlOOH before and after Pb(II) adsorption with several adsorption peaks. The peaks at 3446 cm$^{-1}$ are the hydroxyl group peak of AlOOH and the adsorbed moisture. The peaks that appeared around the wavelength 3059–2851 cm$^{-1}$ are the aromatic C-H stretching vibrations of PS [22]. The aromatic C=C stretching vibration for PS is recorded at 1601, 1492, and 1453 cm$^{-1}$, while the C-H bending vibrations are observed at 750 and 698 cm$^{-1}$. The peak for the AlO-OH showed absorbance at 1384, 1027, and 755 cm$^{-1}$ [23]. A peak appeared at 1154 cm$^{-1}$, belonging to the O-H bending in AlO-OH. Moreover, the peaks appearing in the lower wavelength region belong to the Al-O groups in AlOOH [24]. As shown in Figure 6b, the AlO-OH bands shifted after the Pb(II) adsorption, indicating the interaction between the AlO-OH and Pb(II). These results revealed that hydroxyl groups were the leading active site on the PS/AlOOH hybrid, which is involved in Pb(II) adsorption [25].

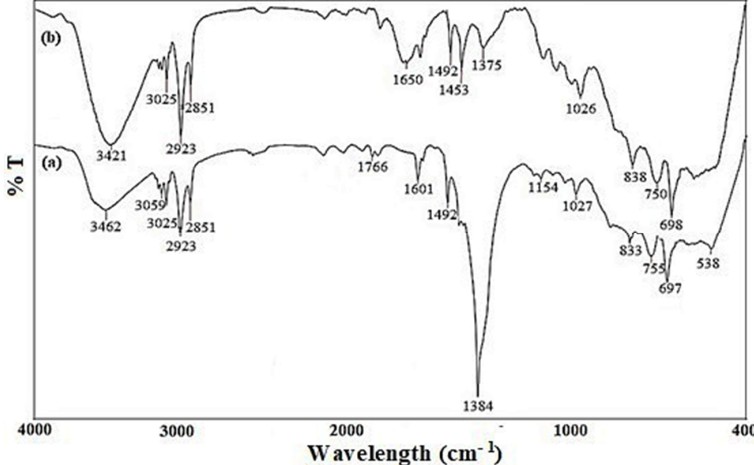

**Figure 6.** FTIR spectrum of the PS/AlOOH (**a**) before adsorption and (**b**) after Pb(II) adsorption.

### 3.2. Adsorption Kinetics

The rate of Pb(II) uptake from the solution and diffusion to the PS/AlOOH surface is generally explored with the kinetics of the adsorption process. The Pb(II) adsorption rate onto the PS/AlOOH hybrid adsorbent as a function of the interaction time is presented in Figure 7. The results in Figure 7 show that the adsorption of Pb(II) onto PS/AlOOH was fast during the initial few minutes due to the empty adsorption sites on the hybrid polymer. The active sites become occupied with the Pb(II) ions as the interaction time increases, and the complete saturation of PS/AlOOH active sites occurred in 90 min, indicating the equilibrium establishment and that no place would remain vacant on the PS/AlOOH [26].

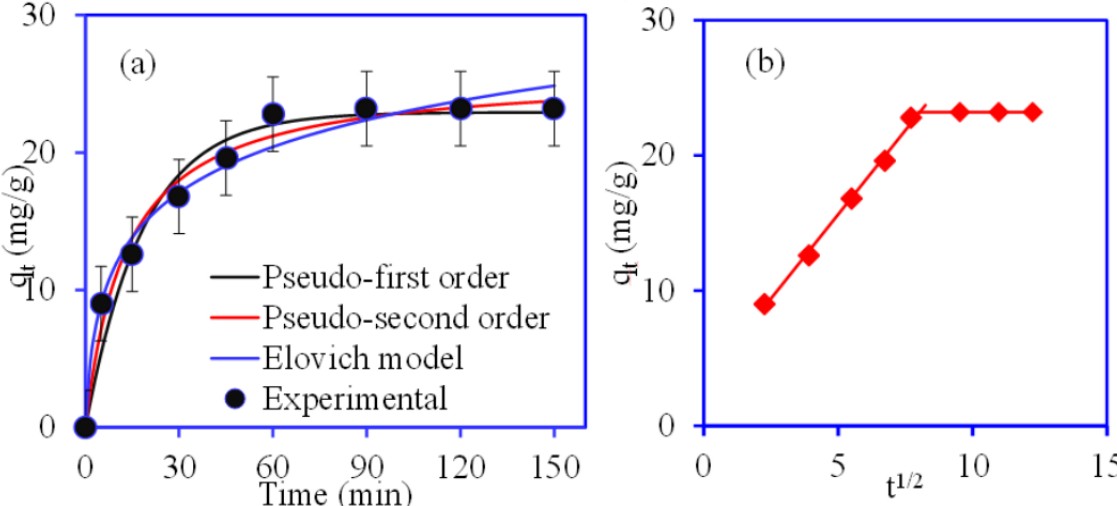

**Figure 7.** (**a**) Adsorption kinetic plots for Pb(II) removal; (**b**) intra-particle diffusion plot (pH—5.6, solution volume—20 mL, Pb(II) conc.—100 mg/L, temp.—32 °C, PS/AlOOH mass—0.05 g).

The Pb(II) adsorption rate onto PS/AlOOH can be investigated by fitting the equilibrium data to non-linear kinetics equations such as pseudo-first-order, pseudo-second-order, and Elovich models. The equation for the Lagergren pseudo-first-order model [27] is as follows:

$$q_t = q_e \left( 1 - e^{(-k_1 t)} \right) \tag{2}$$

where $q_e$ and $q_t$ are the Pb(II) uptake capacities onto the PS/AlOOH (mg g$^{-1}$) at saturation and at time $t$, respectively, while the first-order kinetic model rate constant is represented by $k_1$ min$^{-1}$. The values of the pseudo-first-order parameters were obtained from the plot of $q_t$ against $t$ (Figure 7).

The Pb(II) equilibrium data were also fitted to the non-linear pseudo-second-order model [28], which can be stated as:

$$q_t = \frac{q_e^2 k_2 t}{[k_2(q_e)t + 1]} \tag{3}$$

where $k_2$ (g mg$^{-1}$ min$^{-1}$) is the pseudo-second-order rate constant.

The adsorption kinetic data for Pb(II) removal were also fitted to the Elovich kinetics model, and the non-linear equation for the applied model is as follows:

$$q_t = \frac{1}{\beta} \ln \alpha \beta_t \tag{4}$$

where $\alpha$ and $\beta$ are the Elovich model rate constants.

The analysis of the best-fitted model was evaluated using the error function assessment by measuring the difference between the theoretical and experimental data. The validation of the applied kinetic models was assessed by fitting the root-mean-square error (RMSE)

and chi-square ($\chi^2$) statistical error functions. The equations for the applied statistical error functions are as follows:

$$RSME = \sqrt{\frac{1}{N-M} \sum_{i=1}^{N}(q_e exp - q_e cal)^2} \qquad (5)$$

$$X^2 = \sum_{i=1}^{N}\left[\frac{(q_e exp - q_e cal)^2}{q_e cal}\right] \qquad (6)$$

where $q_e exp$ and $q_e cal$ are the experimental and theoretical Pb(II) uptake capacity (mg g$^{-1}$).

The non-linear fitting of the experimental data to the pseudo-first-order, pseudo-second-order, and Elovich models is displayed in Figure 7a. Kinetics parameters and statistical error values are included in Table 1. The superior values of the $R^2$ indicate the fitting of all applied kinetics models to the data [29]. However, the values of the RMSE and $\chi^2$ are the lowest for the pseudo-second-order kinetic model. Moreover, the slightly greater value of the $R^2$ for the pseudo-second-order model indicates that the pseudo-second-order model fits the experimentally investigated data more than the pseudo-first-order and Elovich models. The pseudo-second-order model was best suited to the Pb(II) adsorption data, indicating that the Pb(II) adsorption rate-limiting step is chemisorption via the exchange or/and sharing of the electrons between the metal ions and PS/AlOOH, surface chelation, or ion-exchange. Similar Pb(II) adsorption behavior was also reported on allophane [26] and polystyrene microplastics adsorbent [18].

**Table 1.** The values of kinetic parameters for the adsorption of Pb(II).

| Pseudo-First-Order | | Pseudo-Second-Order | | Elovich Model | |
|---|---|---|---|---|---|
| $q_e$ (mg g$^{-1}$) | 22.934 | $q_e$ (mg g$^{-1}$) | 25.813 | $a$ (mg g$^{-1}$ min$^{-1/2}$) | 5.192 |
| $k_1$ (min$^{-1}$) | 0.0541 | $k_2$ (g mg$^{-1}$ min$^{-1}$) | 0.0029 | $\beta$ (mg g$^{-1}$) | 0.204 |
| $R^2$ | 0.9544 | $R^2$ | 0.9815 | $R^2$ | 0.9711 |
| RMSE | 1.414 | RMSE | 1.040 | RMSE | 1.125 |
| $\chi^2$ | 2.605 | $\chi^2$ | 0.820 | $\chi^2$ | 0.567 |

Since the pseudo-first-order, pseudo-second-order, and Elovich kinetic models provide details only about the rate of Pb(II) adsorption, the Weber and Morris [30] intra-particle mass transfer diffusion model was used to find the distribution of Pb(II) ions on/in the PS/AlOOH surface. The linear equation for the intra-particle diffusion model is as follows

$$q_t = k_{id}\, t^{1/2} + C \qquad (7)$$

where C (mg g$^{-1}$) and $k_{id}$ (mg(g min)$^{-1}$) represent the intercept and intra-particle diffusion rate constant. A plot of $q_t$ versus $t^{1/2}$ is used to find values of the C and $k_{id}$. The $q_t$ vs. $t^{1/2}$ (Figure 7b) demonstrates the two straight-line portions. The first part of the curves signifies the boundary layer distribution of the Pb(II) ions on the PS/AlOOH surface, whereas the second line results from the intra-particle diffusion. The intercept value details the distribution of Pb(II) onto the PS/AlOOH surface. The higher intercept values lead to boundary layer adsorption. The intercept value for Pb(II) is 16.33 mg g$^{-1}$. The result designated that the scavenging of Pb(II) onto PS/AlOOH was controlled through a boundary layer effect [17,18].

*3.3. Effect of Solution pH*

The scavenging of Pb(II) onto PS/AlOOH was performed at pH 2.0 to 6.0 to investigate the connection between solution pH and uptake capacity. The pH study results shown in Figure 8 indicate that the adsorption of Pb(II) improved with the rise in the solution pH, and the highest Pb(II) adsorption onto PS/AlOOH was noted at pH 6. The adsorbed Pb(II) amount increased rapidly from pH 2.0 to 4.0, and, after that, very slow adsorption was

observed. In a highly acidic medium, Pb(II) scavenging was low due to competition between positively charged excess H$^+$ and Pb(II) toward the same binding sites of PS/AlOOH. With the increasing solution pH, the negative charge density on the PS/AlOOH surface increases, which favors the binding of positively charged Pb(II) ions [31,32]. The possible ion exchange and complexation reactions on the surface of PS/AlOOH may occur as follows:

$$AlO–OH + H \rightarrow AlO–OH_2^{+ \text{(low pH)}}$$
$$AlO–OH + Pb(II) \rightarrow (AlO–O–Pb)^+ + H^+$$
$$AlO–O^- + Pb(II) \rightarrow (AlO–O–Pb)^+$$
$$AlO–OH + Pb(II) + H_2O \rightarrow (AlO–O–PbOH) + 2H^+$$

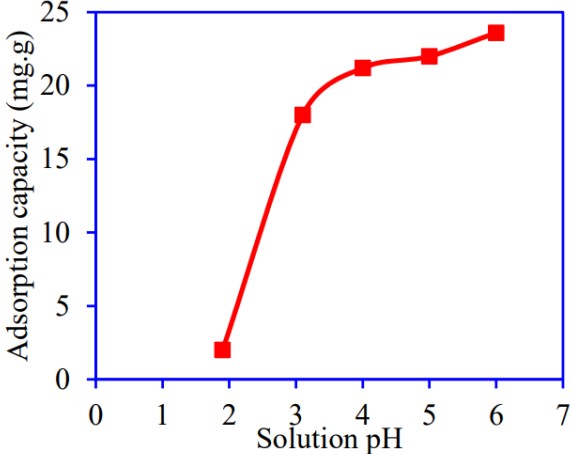

**Figure 8.** Adsorption of Pb(II) at varying solution pH values (time—150 min, solution volume—20 mL, Pb(II) conc.—100 mg/L, temp.—32 °C, PS/AlOOH mass—0.05 g).

### 3.4. Effect of Concentration and Adsorption Isotherm

The effect of Pb(II) concentrations in the solution on its scavenging onto PS/AlOOH was investigated in ranges from 20 to 100 mg L$^{-1}$ to ensure the adsorption equilibrium concentration. As seen in Figure 9, the adsorption improved from 7.6 to 23.2 mg g$^{-1}$ with the increased metal ion concentration. The adsorption capacity of Pb(II) onto the PS/AlOOH polymeric hybrid increased with the rise in the initial Pb(II) concentration, which might be attributed to an increase in the concentration gradient driving force with the rise in the initial Pb(II) ion concentration [33].

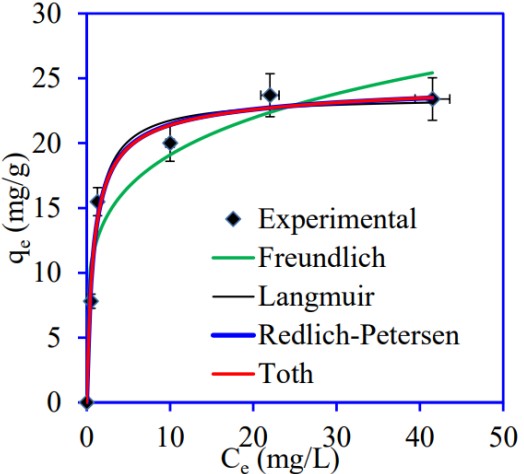

**Figure 9.** Adsorption isotherm models plot for Pb(II) removal (pH—5.6, solution volume—20 mL, temp.—32 °C, PS/AlOOH mass—0.05 g).

Langmuir, Freundlich, Redlich–Peterson, and Toth adsorption isotherms were applied to find the PS/AlOOH hybrid adsorbent's adsorption behaviors and surface properties. The non-linear equations for the Langmuir, Freundlich, Redlich–Peterson (R–P), and Toth isotherm models, respectively, can be expressed as follows:

$$q_e = \frac{q_m k_L C_e}{1 + k_L C_e} \tag{8}$$

$$q_e = k_F C_e^{\frac{1}{n}} \tag{9}$$

$$q_e = \frac{K_{RP} C_e}{1 + \alpha_{RP} C_e^{\beta}} \tag{10}$$

$$q_e = \frac{q_m C_e}{\left(K_{To} + C_e^{tn}\right)^{1/tn}} \tag{11}$$

where $C_e$: Pb(II) at equilibrium concentration, $q_e$: Pb(II) at equilibrium capacity, $q_m$: monolayer adsorption capacity, $k_F$ and $1/n$: Freundlich constant associated with adsorption intensity and uptake capacity, $K_{RP}$ (L/g) and $\alpha_{RP}$: Redlich–Peterson isotherm constants, and β: exponent reflecting the heterogeneity of the PS/AlOOH hybrid adsorbent. $K_{TO}$ and $tn$ are the Toth model constant (dimensionless) and exponent ($0 < n \leq 1$), respectively. The non-linear plots for the applied isotherm models are included in Figure 9, and the obtained isotherm parameters and their corresponding error function analysis RMSE and $\chi^2$ are included in Table 2. The values of the $R^2$, RMSE, and $\chi^2$ are very close for the applied models. However, the value of $R^2$ for the Toth model is slightly higher than that for the Langmuir, Freundlich, and R–P isotherm models. Upon further comparison of the error function RMSE and $\chi^2$ values, Toth gave the lowest values, while the Langmuir model showed the lowest $\chi^2$ values. Empirically, the Toth model is the modified Langmuir equation, which suggests the heterogeneous adsorption of Pb(II) onto the PS/AlOOH hybrid adsorbent [34]. The values of $R^2$, RMSE, and $\chi^2$ indicate that the Toth isotherm model predicts the scavenging of Pb(II) onto the PS/AlOOH hybrid adsorbent through the chemisorption.

**Table 2.** The values of isotherm parameters for Pb(II) adsorption onto the PS/AlOOH adsorbent.

| Langmuir | | Freundlich | | Redlich–Petersen | | Toth | |
|---|---|---|---|---|---|---|---|
| $q_m$ | 23.611 mg g$^{-1}$ | n | 4.962 | $K_{RP}$ | 30.555 L g$^{-1}$ | $q_m$ | 21.355 mg g$^{-1}$ |
| $K_L$ | 1.168 L mg$^{-1}$ | $K_f$ | 2.003 mg g$^{-1}$ (mg L$^{-1}$)$^{-1/n}$ | $a_{RP}$ | 1.412 L mg$^{-1}$ | $K_{To}$ | 0.699 L mg$^{-1}$ |
| | | | | β | 0.972 | $tn$ | 0.984 |
| $R^2$ | 0.9760 | $R^2$ | 0.9438 | $R^2$ | 0.9863 | $R^2$ | 0.9864 |
| RMSE | 1.036 | RMSE | 2.047 | RMSE | 1.009 | RMSE | 1.004 |
| $\chi^2$ | 0.390 | $\chi^2$ | 3.457 | $\chi^2$ | 0.411 | $\chi^2$ | 0.409 |

### 3.5. Pb(II) Adsorption Thermodynamics

The impact of temperatures on the scavenging of Pb(II) from the aqueous solution onto PS/AlOOH was performed at a temperature range from 305 to 323 K, as shown in Figure 10a. It was noticed that the uptake capacity of PS/AlOOH enhanced very slight from 23.2 to 24 mg g$^{-1}$ with the increase in the Pb(II) solution temperature. The reduction in solution viscosity and the increase in the Pb(II) ions mobility may cause a slight increase in the metal ions scavenging. Moreover, the deformation of the active sites and diffusion of Pb(II) ions may be increased in PS/ALOOH at higher temperatures. Furthermore, ions are readily dissolved at a higher temperature, so their adsorption becomes more favorable [35].

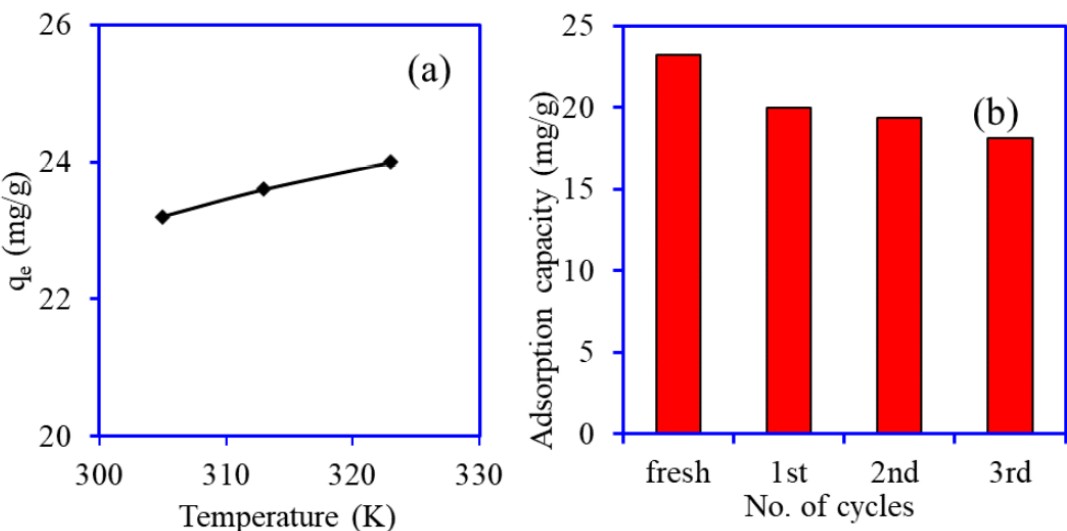

**Figure 10.** (**a**) Adsorption of Pb(II) onto PS/AlOOH at varying solution temperatures, and (**b**) reusability study of PS/AlOOH.

Thermodynamic examinations were performed to observe the energetic changes related to Pb(II) adsorption onto PS/AlOOH. The following were applied to the thermo-dynamic data to obtain the values of standard free energy change ($\Delta G^\circ$), enthalpy change ($\Delta H^\circ$), and entropy change ($\Delta S^\circ$):

$$\Delta G^\circ = -RT \ln K_c$$

$$\ln K_c = (\Delta S^\circ/R) - (\Delta H^\circ/RT)$$

where, $K_c$, T, and R are the distribution coefficient, temperature (K), and gas constant (8.314 J mol$^{-1}$ K$^{-1}$), respectively. The thermodynamic parameters ($\Delta H^\circ$ and $\Delta S^\circ$) values were determined from the slope and intercept of the ln Kc vs. 1/T plot, and the values are presented in Table 3. The adsorption of Pb(II) from the aqueous solution onto PS/AlOOH shows negative $\Delta G^\circ$ values, implying that Pb(II) scavenging was spontaneous. The positive value of $\Delta H^\circ$ confirmed that adsorption was endothermic in nature. Furthermore, Table 3, demonstrating a positive value for $\Delta S^\circ$, specifies that the degrees of freedom enhanced at the solid–liquid interface during Pb(II) adsorption onto PS/AlOOH [36].

**Table 3.** Thermodynamic parameters for Pb(II) adsorption on PS/AlOOH.

| Temperature | $-\Delta G^\circ$ (kJ mol$^{-1}$) | $\Delta H^\circ$ (kJ mol$^{-1}$) | $\Delta S^\circ$ (Jmol$^{-1}$ K$^{-1}$) |
|:---:|:---:|:---:|:---:|
| 305 | 0.818 | | |
| 313 | 0.947 | 3.753 | 14.998 |
| 323 | 1.088 | | |

### 3.6. Desorption and Reusability Studies

The Pb(II) desorption from the exhausted PS/AlOOH has been investigated using pure de-ionized water, 0.1M NaCl, and 0.1M HCl. The results demonstrated that de-ionized water and 0.1M NaCl could not break the bonds between Pb(II) and PS/AlOOH, while 0.1M HCl liberated the Pb(II) in the solution. The desorption of the Pb(II) in HCl is due to the ion-exchange process between the H$^+$ ions and Pb(II) [37]. Therefore, 0.1M HCl was used to regenerate the used PS/AlOOH. The results revealed that the uptake capacity of the PS/AlOOH changed from 23.2 mg/g to 18 mg/g after three cycles of reuse, as displayed in Figure 10b. The decline in the uptake capacity might be due to the deactivation of the same active sites and undesorbed Pb(II) ions from the pore of the PS/AlOOH.

### 3.7. Comparison of Adsorption Performance

The adsorption capacities of various adsorbents have been compared with the PS/AlOOH used for decontaminating Pb(II) from an aqueous medium. Table 4 summarizes the adsorption capacities of multiple adsorbents used to scavenge Pb(II). Comparing the adsorption capacity shows that PS/AlOOH is a moderate hybrid adsorbent material for Pb(II) sequestration from an aqueous solution.

**Table 4.** Pb(II) adsorption capacity of various adsorbents.

| Adsorbent | Adsorption Capacity (mg/g) | Ref |
|---|---|---|
| Alumina | 13.11 | [1] |
| Pinecone-activated carbon | 27.53 | [38] |
| Activated carbon | 17.77 | [39] |
| Modified soda lignin | 46.72 | [40] |
| Polypropylene–clinoptilolite | 1.01 | [41] |
| Activated carbon from waste rubber tire | 9.6805 | [42] |
| Biochar | 0.44 | [43] |
| Polypyrrole/oMWCNT | 26.32 | [44] |
| Acidified CNTs | 17.44 | [45] |
| Polypyrrole-based activated carbon | 50 | [46] |
| Acid-activated carbon | 30.3 | [47] |
| PS/AlOOH | 23.61 | This study |

### 4. Conclusions

A facile strategy has been used herein to synthesize the PS/AlOOH hybrid adsorbent. The prepared PS/AlOOH hybrid adsorbent has been successfully utilized for the sequestration of Pb(II) from an aquatic medium. The XRD, EDX, and elemental mapping analysis evidence the successful binding of Pb(II) onto the PS/AlOOH hybrid adsorbent. The findings demonstrated the fast scavenging of Pb(II) onto the PS/AlOOH hybrid within 90 min, and the most significant adsorption was observed at pH 6. The Pb(II) adsorption kinetic data were best suited for the pseudo-second-order kinetic model, and inter-particle diffusion was the rate-limiting step. The Pb(II) adsorption equilibrium data were applied to the Langmuir, Freundlich, Redlich–Petersen, and Toth isotherm models, but the Toth isotherm model was suited. The maximum monolayer adsorption capacity was 23.611 mg g$^{-1}$. The kinetic and isotherm results revealed that the chemisorption and the ion-exchanged mechanism were involved in the Pb(II) scavenging. The thermodynamic study demonstrated that Pb(II) removal by the PS/AlOOH hybrid was not affected much by the change in the solution temperature. Regeneration studies proved that HCl was the best eluent to desorb the Pb(II) from the saturated PS/AlOOH hybrid, and after three times of reuse, the PS/AlOOH hybrid showed a reasonable adsorption capacity. Based on the findings, the PS/AlOOH hybrid can be used as an adsorbent for scavenging pollutants from wastewater.

**Funding:** This research received no external funding.

**Data Availability Statement:** All the data included in the article or available on demand.

**Acknowledgments:** Author gratefully acknowledge the support from the Deanship of Scientific Research and Department of Environmental Sciences, King Abdulaziz University, Jeddah, Saudi Arabia.

**Conflicts of Interest:** The author declares no conflict of interest.

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
