# Peer review of "Fabrication of Polystyrene/AlOOH Hybrid Material for Pb(II) Decontamination from Wastewater: Isotherm, Kinetic, and Thermodynamic Studies"

_colloids, doi:10.3390/colloids6040072_

Round 1

Reviewer 1 Report

Recommendation: This paper is publishable

The authors have examined on “Fabrication of Polystyrene/AlOOH Hybrid Materials for Pb(II) Decontamination from Wastewater: Isotherm, Kinetic, and Thermodynamics Studies”. The author has designed the hybrid materials, characterized them using various techniques, and made valuable discussion which could be applicable for removal of pollutants. At this stage I would agree to publish this manuscript in Colloid and Interfaces.

Nevertheless, I would ask the author to make some minor correction in their manuscript. I have rationalized my thoughts below.

1.      Line 64: there is an extra space between ion exchange and chelation. It should be eliminated.

2.      Line 113: The extra space should be reduced.

3.      Figure 5 (f): The author represents the mixed elements; what does it represent here? Is the overlap of the defined elements in (b), (c), (d), and (e) or different elements?  Clarify?

4.       FTIR analysis:  In the discussion (line 184 to 197), the author should correct the typographical errors to representing the wavenumber (cm-1) for every peaks.  

5.       Line 152: The peak appeared at 1152 cm-1 belong to the O-H bending in AlOOH. I guess it would be 1154 cm-1.

6.      Line 298 and 299: Equation numbers (10 and 11) should be written properly.

7.      Reference [47]: Why the text is in bold?

Author Response

Dear Reviewer

Thanks for your valuable comments and suggestions. Responses to comments is mentioned in the attached file.

Reviewer 2 Report

The manuscript “Fabrication of Polystyrene/AlOOH Hybrid Material for Pb(II) Decontamination from Wastewater: Isotherm, Kinetic, and Thermodynamic Studies” deals with the preparation of a polystyrene/AlOOH (PS/AlOOH) hybrid adsorbent, immobilizing AlOOH nanoparticles in the polystyrene matrix. The application of the PS/AlOOH hybrid was investigated for the scavenging of Pb(II) from an aqueous medium. Generally speaking, the work is interesting and several characterizations of the samples were carried out. However, some revisions are required before the publication, as follows:

- Abstract. Add the percentage of scavenging of the produced device.

- Introduction. Enlarge the state of the art; for this purpose, see for instance these recent works: Somma et al., ChemEngineering, 2021, 5(3), 47; Abdelkhalek et al., Scientific Reports, 2022, 12(1), 7060; Akartasse et al., Polymers, 2022, 14(20), 4265; etc.

- Materials. Add other physico-chemical properties of the used chemicals.

- Methods. The preparation method used to obtain polystyrene/AlOOH hybrid has not any reference, as well as the further characterization methods.

- Results. The description and discussion of the results are poor. A comparison with the previous literature is missing. The relevance of the present findings is not highlighted.

Author Response

Dear Reviewer

Thanks for your valuable comments and suggestions. Responses to comments are mentioned in the attached file.

Round 2

Reviewer 2 Report

The manuscript has been improved.